# Decoupling of brain function from structure reveals regional behavioral specialization in humans

Maria Giulia Preti [1,2]* & Dimitri Van De Ville [1,2]

The brain is an assembly of neuronal populations interconnected by structural pathways. Brain activity is expressed on and constrained by this substrate. Therefore, statistical dependencies between functional signals in directly connected areas can be expected higher. However, the degree to which brain function is bound by the underlying wiring diagram remains a complex question that has been only partially answered. Here, we introduce the structural-decoupling index to quantify the coupling strength between structure and function, and we reveal a macroscale gradient from brain regions more strongly coupled, to regions more strongly decoupled, than expected by realistic surrogate data. This gradient spans behavioral domains from lower-level sensory function to high-level cognitive ones and shows for the first time that the strength of structure-function coupling is spatially varying in line with evidence derived from other modalities, such as functional connectivity, gene expression, microstructural properties and temporal hierarchy.

[1] Institute of Bioengineering, Center for Neuroprosthetics, École Polytechnique Fédérale de Lausanne (EPFL), Campus Biotech, Chemin des Mines 9, 1202 Geneva, Switzerland. [2] Department of Radiology and Medical Informatics, University of Geneva, Campus Biotech, Chemin des Mines 9, 1202 Geneva, Switzerland. *email: maria.preti@epfl.ch

Brain activity is constrained by the anatomical substrate on which it manifests, but how functional activity is shaped by the underlying structural connectivity (SC) remains a central question in neuroscience[1]. Whole-brain imaging techniques such as diffusion-weighted and functional magnetic resonance imaging (fMRI) made it possible to obtain systems-level measures of SC, revealing white-matter pathways, and functional connectivity (FC), reflecting statistical interdependencies of activation time courses, respectively. Several methods have been proposed to relate these measures. First, the link between SC and FC has been most commonly investigated using simple and direct correlational approaches[2,3]. Second, effective connectivity by dynamic causal modeling has explored how neurobiologically plausible models can explain functional signals in terms of excitatory and inhibitory interactions[4], potentially incorporating priors on SC[5]. Third, graph modeling has motivated a broad range of studies that summarize organizational principles of SC or FC[6–8], allowing to extract systems-level network properties of architecture, evolution, development, and alterations by disease or disorder[9,10]. Finally, in order to probe the causal influence of SC on FC, simulation of functional activity starting from the structural connectome and regional neural models of local dynamics have been proposed[11–16]. Recent concepts of network controllability have also looked into how specific empirical patterns of spatial activity can be driven through the structural connectome[17]. Lately, structural brain networks have been explored with graph harmonic analysis, a powerful approach that relates to fundamental concepts such as Laplacian embedding[18] and spectral clustering[19]. Basically, the product of all local connectivity is summarized in harmonic components—spatial patterns defined on the nodes of the graph—that reveal global network organization. These components showed to be reminiscent of functional resting-state networks[20]. The next advance was to decompose functional signals in terms of these structurally informed components[21], opening new avenues to look into the relationship between structure and function[22,23].

Here, we exploit this framework to provide insights into how brain function couples with structure. The term *coupling* refers to the dependency of functional signals on the anatomical structure as measured by functional signal smoothness on the structural graph, not to be confused by its meaning in other fields of neuroscience (e.g., neuromodulation). First, we investigate the coupling strength across the brain, by defining a filtering operation that splits brain activity at every moment in time in two parts with, on average, equal energy: one that is weakly coupled with structure and the other one strongly so. Their energy ratio then leads to the *structural-decoupling index*, which can be determined per brain region. Second, we deploy a new non-parametric test to assess the significance of the structural-decoupling index, based on a strong null model that maintains selected properties of the interplay between functional activity and structural connectome. Using data from the Human Connectome Project (HCP), we find that activity in sensory regions, including visual, auditory, and somatomotor, is more strongly coupled with structure, while the opposite is true for higher-level cognitive regions such as parietal (executive control networks), temporal (amygdala, language area), orbitofrontal ones. Third, we rank brain regions by their structural-decoupling index and explore their behavioral relevance using a meta-analysis of the literature associating specific topic terms to brain areas. This shows that characterizing brain areas based on their structure–function relation reveals a macroscale organization of the cortex placing at one side (low structural-decoupling index) areas related to lower-level functions (sensory, motor), while at the other (high decoupling) more complex functions (e.g., memory, reward, emotion). These findings turn out to be highly reliable in terms of test–retest analysis.

## Results

**Harmonics of the structural connectome.** The structural connectome (Fig. 1a) can be modeled as a graph from which the harmonic components can be computed by the eigendecomposition of the Laplacian. Harmonic components, as illustrated in Fig. 1b, are graph signals—values associated to nodes—that maximally preserve distances on the graph. Therefore, they provide a natural spectral representation of any graph signal in terms of increasing complexity, which corresponds to the notion of frequency. To confirm this intuition, Supplementary Fig. 1 reports the weighted zero crossings along the graph structure for each harmonic component. Low-frequency ones (examples shown in Fig. 1b) capture brain patterns of global and slow variations along the main geometrical axes (e.g., anterior–posterior, left–right), while higher frequencies encode increasingly complex and localized patterns. This type of cortical decomposition is similar to results obtained with the same technique on different parcellations[20], as well as with different approaches; e.g., neural field theory[24,25].

**Brain activity couples with the structural connectome.** Resting-state activity is then projected on the structural-connectome harmonics[21–23]; i.e., for each timepoint, the spatial pattern of activation is represented as a weighted linear combination of harmonic components (Fig. 1c). The time-averaged squared weights form the energy spectral density of the resting-state activity, as shown in Fig. 1c (inset) and Supplementary Fig. 2. One can notice that brain activity is expressed preferentially by lower-frequency components, following a trend that is reminiscent of power-law behavior.

**Null models of brain activity informed by the structure.** We generate two types of surrogate functional data—with and without knowledge of the brain SC, respectively—by randomizing, for each timepoint, the signs of the empirical graph spectral coefficients[26]. The surrogate activation pattern is then obtained by reconstruction. In the first case (*SC-ignorant surrogates*), no information about the empirical SC is incorporated in the model, as the functional signals are projected onto harmonics of an artificially generated graph which just preserves the degree of the original SC. In the second case (*SC-informed surrogates*), instead, the coefficients of empirical structural harmonics are permuted and used for reconstruction, obtaining functional signals that are randomized, but built on top of the real structural architecture. This surrogate method is inspired by statistical resampling that generally operates in the temporal or spatial domain through shuffling of Fourier[27] or wavelet[28,29] coefficients. The rationale is to preserve a meaningful feature of the empirical data (e.g., the energy distribution of Fourier or wavelet coefficients), thus leading to surrogates that embrace a strong null hypothesis. By design, our surrogate data preserve the empirical energy spectral density, but specific interactions between harmonic components as expressed by empirical activity are destroyed. Since the same sign randomization is applied to each timepoint, correlations between timepoints and non-stationarities are maintained. The procedure can be repeated multiple times to obtain a set of surrogates from which null distributions of test metrics can be derived.

We first use these null models to observe how FC appears, once the empirical structural and/or functional features are removed from the data. Fig. 1d shows that, as one might expect, SC-ignorant surrogate FC does not display any particular pattern. Conversely, SC-informed surrogate FC highlights structured patterns that are reminiscent of the underlying SC; nodal strength of regions in occipital gyri, cuneus, precuneus, frontal, pre- and

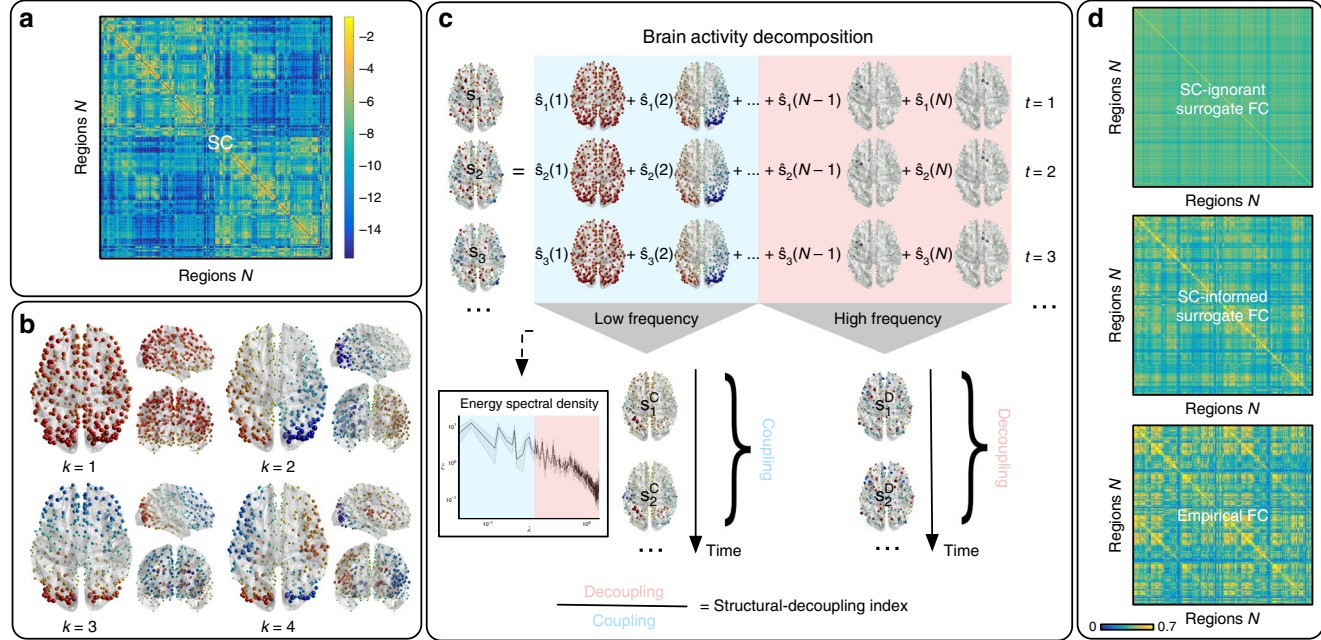

**Fig. 1** Method pipeline. **a** Structural connectome (SC) between $N = 360$ atlas regions, displayed in logarithmic scale. **b** SC eigendecomposition leads to structural harmonics with increasing spatial frequency $k$. **c** Brain activity at every time point $t$ ($\mathbf{s}_t$) is written as a linear combination of harmonics (by using coefficients $\hat{\mathbf{s}}_t$). The median-split criterium on the activity energy spectral density $\xi$ (inset) is used to split the spectrum and decompose brain activity into coupled/decoupled portions $\mathbf{s}_t^C$ and $\mathbf{s}_t^D$ (using low/high-frequency harmonics, respectively; $\lambda$ = harmonic frequency). The ratio between decoupled/coupled signal norms is defined as structural-decoupling index. **d** Surrogate functional signals are generated with/without knowledge of SC, by spectral coefficient randomization. Average functional connectomes (FC) obtained from correlating pairs of empirical/surrogate functional signals are compared

post-central and inferior parietal gyri, are standing out. However, one can observe visually that the connectivity patterns of empirical FC time courses appear more contrasted than the ones of SC-informed surrogates. To evaluate the additional information content present in empirical FC, we compare the nodal strengths of SC-informed surrogate and empirical FC matrices with the ones of the structural connectome. Notably, the Spearman correlation between surrogate FC and SC is significantly stronger than the one between empirical FC and SC ($r = 0.93$ vs $r = 0.46$, difference significant with $p < 0.05$, assessed with non-parametric test performed across surrogates), confirming that empirical FC is the result of a complex interplay captured by specific sign configurations of major structural components that are represented by the harmonics.

**Brain activity decomposed according to structural coupling**. To investigate the degree of coupling of function with structure, we introduce spectral low- and high-pass windows based on median split on the observed energy spectral density of functional data, shown in Fig. 1c (inset) and in Supplementary Fig. 2. The median-split frequency occurs at $C = 21$, corresponding to $\lambda_C = 0.36$. The functional data are then filtered, timepoint by timepoint, by applying the spectral windows of ideal low- and high-pass filters, respectively, to the harmonic coefficients. The reconstruction can be evaluated per node (brain region) in terms of the ratio of the energies of activity decoupled (high-pass) versus coupled (low-pass) with respect to structure; i.e., the *structural-decoupling index*. The statistical significance of these nodal measures is assessed by comparison with SC-informed surrogates. Figure 2 shows the average structural-decoupling index (in binary logarithm form) for surrogate (generated with or without knowledge of SC) and empirical functional signals. The first distribution (Fig. 2a) displays a high structural decoupling for SC-ignorant surrogates, as expected. The influence of the structure can be seen instead when looking at the coupling of

SC-informed surrogate functional signals (Fig. 2b). In this case, in fact, a pattern resembling the known structural core of connections present in the human brain[3], including posterior medial and parietal cortical areas, shows higher coupling to the structural graph (blue areas). Interestingly, when evaluating the structural decoupling of empirical functional time courses (Fig. 2c), two distinct patterns emerge as significantly more or less decoupled than expected, respectively: the former, mainly including orbitofrontal, temporal, parietal regions, identifying a high-level cognition network (Fig. 2c, red); the latter focused on primary sensory areas, spanning auditory (temporal), visual (occipital), and somatomotor (pre-/post-central) networks (Fig. 2c, blue).

**Structural decoupling reveals behaviorally relevant gradient**. A NeuroSynth meta-analysis based on the same 24 topic terms as implemented by Margulies et al.[30] was applied to the gradient defined by the structural-decoupling index. As shown in Fig. 3a, this reveals a spectrum of macroscale cortical organization that associates structurally coupled regions with multisensory processing, visual perception, motor/eye movements, auditory processing, on one end, and structurally decoupled regions with reward, emotion, affective processing, social cognition, verbal/visual semantics, memory, cognitive control, on the other end. Intriguingly, this result is consistent with previous findings that were based on a gradient defined by FC only[30] (Fig. 3b).

**Reliability across sessions**. The analysis across the two test–retest resting-state acquisitions of the same subjects revealed a very high reliability of the results, reported in Supplementary Figs. 4 and 5 for the second dataset. The median-split cut-off frequency of the Laplacian eigenvalues spectrum for the second dataset occurred at $C = 22$. The spatial correlation between the patterns of structural-decoupling index in the two acquisitions was 0.90 for the empirical case (Fig. 2c, Supplementary Fig. 4c), and 0.99 for

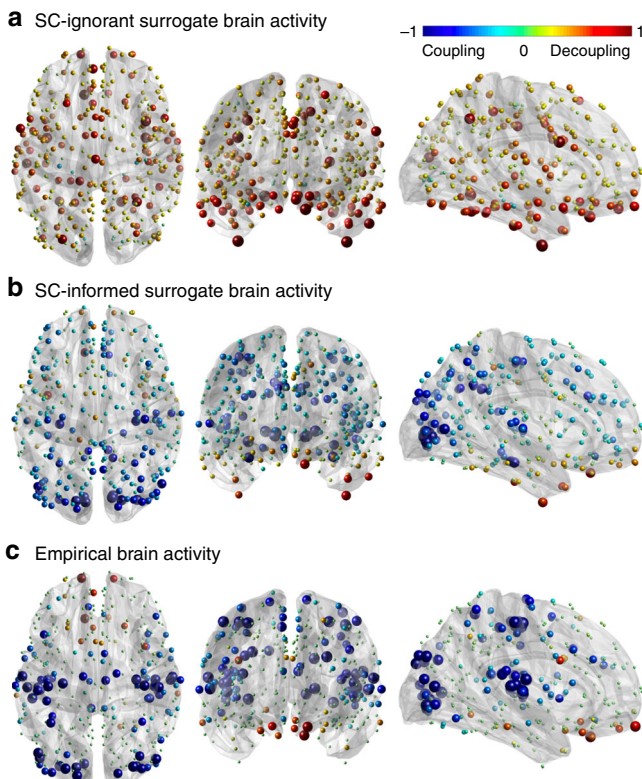

**a** SC-ignorant surrogate brain activity

−1 ▁▁▁▁▁ 1
Coupling    0    Decoupling

**b** SC-informed surrogate brain activity

**c** Empirical brain activity

**Fig. 2** Structural-decoupling index as a new measure of regional coupling between function and structure. The binary logarithm of the index is plotted here for three different brain activity signals, highlighting their coupling to the structural connectome. As the index is reported in logarithmic scale, a value of 1 indicates a double structural decoupling of a region with respect to coupling; vice versa, a value of −1 corresponds to a double coupling with respect to decoupling. The evaluated brain activity signals are: **a** surrogate brain activity time courses without knowledge of the empirical structural connectome: as expected, their structural-decoupling index shows high decoupling from the structural graph; **b** surrogate brain activity time courses with knowledge of the underlying structural connectome, build as a linear combination of structural harmonics with randomized coefficient signs: the structural-decoupling index shows here a pattern of function-structure coupling purely driven by the structural graph, and, in fact, resembling the known structural core of densely interconnected and topologically central regions, mainly composed of posterior medial and parietal cortical areas[3]; **c** empirical brain activity, and displaying only regions with a structural decoupling significantly different with respect to the surrogates in (**b**). Two main patterns emerge, one with regions whose functional activity significantly couples with the structural connectome, including primary sensory and motor networks (blue), the other composed of regions whose functional signals detach from the structure more than expected, including orbitofrontal, temporal, parietal areas (red). Source data to reproduce this figure are provided as a Source Data file

the surrogates (Fig. 2a, b, Supplementary Fig. 4a, b), where in fact the main influence is given by the structure, which is the same across datasets. The meta-analysis also provided a very similar behavioral characterization of the structural-decoupling gradient in the two cases (Fig. 3a, Supplementary Fig. 5).

## Discussion

Brain activity is naturally shaped by the anatomical backbone[2]; however, the degree to which this happens remains difficult to quantify. Previous simulation approaches, in particular, have proposed large-scale neural population models coupled with SC, to explain some of the patterns of empirical FC[12], including modular organization and spatiotemporal dynamics[31]. Such a generative approach, at the macroscale level, allows to validate properties of brain activity that emerge from interactions between brain regions, given a model for regional dynamics and inter-regional connectivity constraints.

Here, instead, we adhere to an alternative approach where the empirical measures of functional brain activity are kept central. Using the harmonic decomposition of the structural connectome, we first show clear evidence that observed brain activity is preferentially expressed using components with lower graph frequencies; i.e., those that fit "better" to the connectome constraints. The distribution dominated by low frequencies followed by the energy spectral density of functional signals projected onto structural harmonics, in accordance with the findings of Atasoy et al.[21], indicate that activity patterns do preferentially express smoothness on the connectome.

This observation is key to establish the structural-decoupling index, which quantifies the function-structure relationship. Brain activity is first filtered into two parts: one by keeping low-frequency components—coupled with the SC—and the other by keeping high-frequency components—decoupled from the SC. The ratio of the energies of these parts can be then computed and evaluated per brain region. We assess the decoupling index obtained under three scenarios: for surrogate functional data based on a simple configuration model of the empirical SC; for surrogate functional data based on the empirical SC; and for empirical functional data. The key property that makes empirical data stand out is that the use of structurally informed components is not organized randomly; i.e., activation patterns arise with specific combinations of structurally informed components, which are randomized in surrogate data (although the amplitudes are preserved). The surrogates induce a null distribution of the decoupling index and thus allow to detect significant function-structure coupling strength in empirical data. This reveals a macroscale gradient of regions that are ordered from being significantly more coupled with structure to significantly less coupled. This gradient basically opposes coupled sensory–motor regions against decoupled higher-cognitive areas. The meta-analysis confirms that the obtained gradient corresponds to a behaviorally relevant ordering from lower to higher-level cognitive functions, similarly to the cortical organization shown by Margulies et al.[30] based on FC data. These findings further corroborate the large body of evidence arguing for the existence of a global gradient in cortical organization spanning between primary sensorimotor and transmodal regions[32], demonstrated so far not only for FC[30] but also for cortical microstructure[33], gene expression[34,35], and temporal hierarchy[36,37]. In particular, the length of functional processing timescales was reported to vary from milliseconds–seconds for sensory–motor regions, characterized by brief transient activity, to seconds–minutes for transmodal association areas, encoding slower intrinsic dynamics[32,36,37]. In addition, a similar depiction is provided by genetic imaging work, where low- versus high-level regions are characterized by the expression of genes favoring temporal precision of fast-evoked neural transmission versus slower, sustained or rhythmic activation, respectively[35]. The higher coupling strength of sensory–motor areas can thus be motivated by their need for reacting fast and reliably to external (and internal) stimuli. On the contrary, high-level cognitive processes such as episodic memory or self-referential thoughts are less predictable, thus more decoupled from the SC. This interpretation is also corroborated by previous work on fMRI fingerprinting, showing that only high-level regions carry subject-specific information[38].

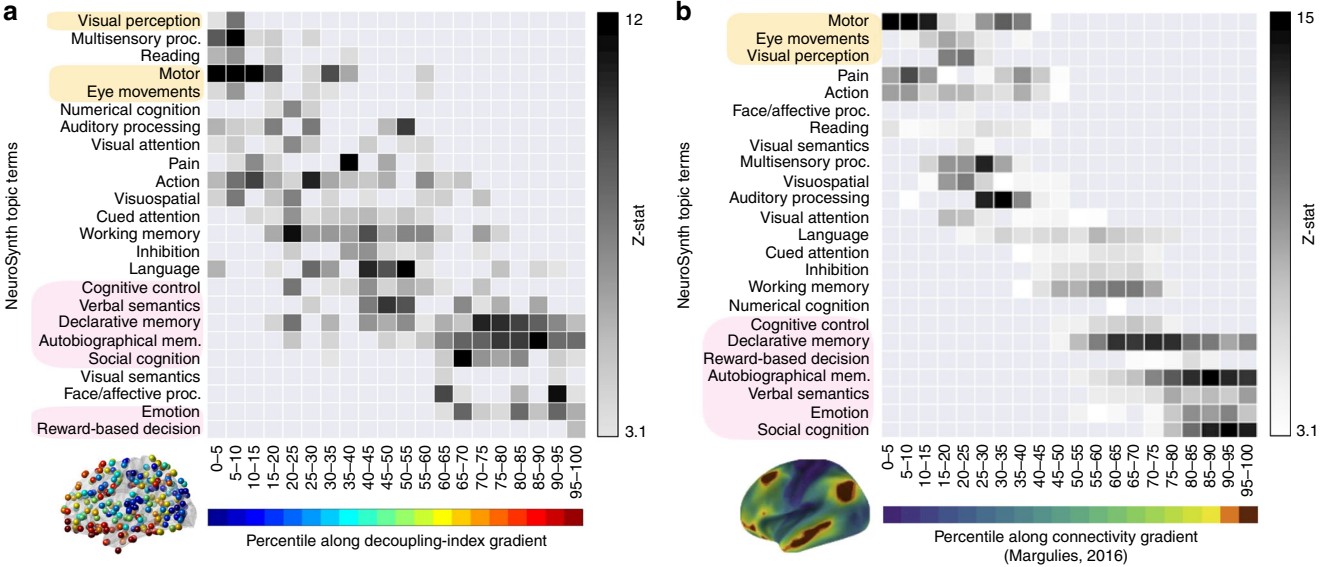

**Fig. 3** Structural-decoupling index reveals organization according to a behaviorally relevant gradient. **a** NeuroSynth meta-analysis is applied to the decoupling index gradient; **b** similar analysis conducted in ref. [30] on a functional connectivity gradient (right). Our analysis shows for the first time that the strength of structure–function coupling orders regions according to behavioral relevance, in accordance with other known principles of brain organization. In fact, despite the two analyses have a different input, a similar trend is found, correlating regions at one extreme of the gradient to lower-level sensory–motor functions and regions at the opposite extreme to higher-cognitive functions. The regions found at the extremes in ref. [30] are highlighted in both diagrams. Source data to reproduce **a** are provided as a Source Data file. **b** Adapted with permission from Fig. 4 in ref. [30]

The visual (occipital) areas are the first ones partially differentiating from the sensory–motor network when we look at the difference between empirical and surrogate structural coupling (Supplementary Fig. 3). This interestingly matches findings from different approaches that report a differentiation of visual regions from other sensory modalities. In particular, a secondary gradient of connectivity, shown by Margulies et al.[30], places the visual cortex at the opposite of somatomotor and auditory regions. Similar findings have been observed in genetic studies, where a secondary genetic expression gradient reveals the same differentiation between sensory modalities[34]. Further, this separation of the visual cortex has been found also in cortical thickness gradients[39], showing higher values for the regions in primary sensory networks except for the occipital areas.

The influence of different properties of the SC graph on the decoupling index could be further investigated. In particular, while the configuration model only kept nodal strengths, another option could be to preserve distances of the spatial embedding, as proposed in ref. [40].

Finally, we note that the proposed framework does not include nor compensate for a noise component. Assuming that the noise contribution is uniform in the spectral domain, the decoupling estimate would be positively biased if noise is ignored. For instance, regions in the orbitomedial prefrontal cortex, which are prone to high susceptibility artifacts, could turn out more decoupled than in reality. Therefore, the meta-analysis is particularly important to provide evidence of the functional relevance of the decoupling index, indicating that it captures patterns that are broadly more meaningful than the reflecting poor signal-to-noise ratio.

In sum, this study demonstrated a principled approach to quantify the coupling strength of functional signals with underlying structure. The methodology opens new avenues of research to investigate inter-regional differences of coupling, as well as intra-regional variations; e.g., over time or experimental conditions. Alterations due to neurological disease and disorder might be another promising application that could lead to new insights.

## Methods

**Overview.** Our approach benefits from the emerging framework of graph signal processing, which revisits classical signal processing operations in the graph setting[41]. For its application to human brain imaging, the normalized graph adjacency matrix **A** is given by the structural connectome, while time-dependent graph signals are taken from the functional data; i.e., activation levels are associated to the nodes of the graph. The eigendecomposition of the graph Laplacian operator $\mathbf{L} = \mathbf{I} - \mathbf{A}$ then provides the harmonic components from which the graph Fourier transform (GFT) can be built. In particular, graph signals can be represented as weighted linear combinations of these components[20,21] and meaningful operations can be introduced (e.g., graph filtering and randomization), which take into account the brain anatomical backbone[23].

**Data.** Fifty-six healthy volunteers from the HCP [db.humanconnectome.org] were included in the study. All experiments were reviewed and approved by the local institutional ethical committee (Swiss Ethics Committee on research involving humans). Informed consent forms, including consent to share de-identified data, were collected for all subjects (within the HCP) and approved by the Washington University institutional review board. All methods were carried out in accordance with relevant guidelines and regulations. The following sequences were used: Structural MRI: 3D MPRAGE T1-weighted, TR = 2400 ms, TE = 2.14 ms, TI = 1000 ms, flip angle = 8°, FOV = 224 × 224, voxel size = 0.7 mm isotropic. Diffusion-weighted MRI: spin-echo EPI, TR = 5520 ms, TE = 89.5 ms, flip angle = 78°, FOV = 208 × 180, 3 shells of $b = 1000, 2000, 3000$ s mm$^{-2}$ with 90 directions plus 6 $b = 0$ acquisitions. Two sessions of 15 min resting-state fMRI: gradient-echo EPI, TR = 720 ms, TE = 33.1 ms, flip angle = 52°, FOV = 208 × 180, voxel size = 2 mm isotropic. Additional two resting-state fMRI sessions (same acquisition parameters) were included in a separate analysis to test for reliability. Two out of the 56 subjects needed to be excluded from this second resting-state dataset for incomplete acquisition. HCP-minimally preprocessed images[42] were used for all acquisitions.

**Structural connectome.** Diffusion-weighted scans were analysed using MRtrix3 [http://www.mrtrix.org/] with the following operations: multi-shell multi-tissue response function estimation, constrained spherical deconvolution, tractogram generation with 10^7 output streamlines. Glasser's multimodal cortical atlas[43] converted to volume was split into the two hemispheres (first 180 areas on the left and last 180 on the right) and used to parcellate the cortex into $N = 360$ regions of interest and generate the structural connectome. The chosen connectivity measure was the number of fibers connecting two regions divided by the region volumes (sum of connected regions). A group connectome $\mathbf{A}_{unnorm}$ was obtained by averaging all subjects' structural matrices. Symmetric normalization led to the adjacency matrix $\mathbf{A} = \mathbf{D}^{-1/2}\mathbf{A}_{unnorm}\mathbf{D}^{-1/2}$ where **D** is the degree matrix.

**Resting-state functional data.** Functional volumes were spatially smoothed with an isotropic Gaussian kernel (5 mm full-width at half-maximum) using SPM8 [https://www.fil.ion.ucl.ac.uk/spm/]. The first 10 volumes were discarded so that the fMRI signal achieves steady-state magnetization, resulting in $T = 1190$ time-points. Individual tissue maps were segmented from the T1 image (white matter, gray matter, cerebrospinal fluid). Voxel fMRI time courses were detrended and nuisance variables were regressed out (6 head motion parameters, average cerebrospinal fluid, and white matter signal). Then, the preprocessed voxel time courses were band-pass filtered $[0.01−0.15 \ \mathrm{Hz}]$ to improve signal-to-noise ratio for typical resting-state fluctuations. Finally, Glasser's multimodal parcellation (the same used for the structural connectome) resliced to fMRI resolution was used to parcellate fMRI volumes and compute regionally averaged fMRI signals. These were z-scored and stored in the $N \times T$ matrix $\mathbf{S} = [\mathbf{s}_t]_{t=1,\dots,T}$. Functional connectivity was computed as Pearson correlation between time courses and averaged across subjects. Node strengths of the functional connectome were assessed as the sum of absolute correlation values for each connection.

**Structural-connectome harmonics.** We defined the GFT by eigendecomposition $\mathbf{LU} = \mathbf{U\Lambda}$ of the graph Laplacian $\mathbf{L}$. The eigenvalues $[\mathbf{\Lambda}]_{k,k} = \lambda_k$ can be interpreted as frequencies, and the eigenmodes $\mathbf{u}_k$ as frequency components, referred to as structural connectome harmonics. Therefore, $\mathbf{u}_k$ with low $\lambda_k$ encode low frequencies and thus smooth signals with respect to the structural network. The GFT converts a graph signal $\mathbf{s}_t$ into its spectral representation $\hat{\mathbf{s}}_t$ and vice versa, by $\hat{\mathbf{s}}_t = \mathbf{U}^\mathrm{T}\mathbf{s}_t$, and $\mathbf{s}_t = \mathbf{U}\hat{\mathbf{s}}_t$. This assumes that all contributions to the energy density are signal relevant.

**Null model generation.** We used spectral randomization[26] to generate two types of surrogate functional signals, $\mathbf{s}^{(\mathrm{rand1})}$ and $\mathbf{s}^{(\mathrm{rand2})}$, ignoring or incorporating knowledge about SC, respectively. This method consists of sign randomization of the graph spectral coefficients, i.e. the harmonics weights, in the reconstruction of surrogate functional signals. For the former case, we used the configuration model to generate a graph $\mathbf{A}'$ preserving the same degree of $\mathbf{A}$, and we used its harmonics $\mathbf{U}'$ for surrogate signal reconstruction, as indicated in Eq. (1):

$$\mathbf{s}_t^{(\mathrm{rand1})} = \mathbf{U}'\hat{\mathbf{s}}_t^{(\mathrm{rand1})} = \mathbf{U}'\mathbf{P}_1\hat{\mathbf{s}}_t' = \mathbf{U}'\mathbf{P}_1\mathbf{U}'^\mathrm{T}\mathbf{s}_t. \quad (1)$$

where $\mathbf{P}_1$ is a diagonal matrix with random $+1/-1$ values. For the latter one, the empirical SC harmonics were used instead, generating the surrogate signals shown in Eq. (2):

$$\mathbf{s}_t^{(\mathrm{rand2})} = \mathbf{U}\hat{\mathbf{s}}_t^{(\mathrm{rand2})} = \mathbf{U}\mathbf{P}_2\hat{\mathbf{s}}_t = \mathbf{U}\mathbf{P}_2\mathbf{U}^\mathrm{T}\mathbf{s}_t. \quad (2)$$

where $\mathbf{P}_2$ is also a diagonal matrix with random $+1/-1$ values. For each considered resting-state session, we generated 19 surrogates. Surrogate FC was computed as Pearson correlation between surrogate time courses and averaged across surrogates and subjects. FC node strengths were assessed as the sum of absolute correlation values for each connection.

**Structural-decoupling index.** Inspired by Medaglia et al.[22], graph signal filtering was implemented in order to decompose the functional signal into one part well coupled with structure (i.e., represented by low-frequency eigenmodes of the graph) and one that is less coupled (i.e., by higher-frequency eigenmodes). This is achieved using the GFT and spectral filtering with an ideal low-pass/high-pass filter. Since the cut-off frequency $C$ is difficult to select, we propose to split the spectrum into two portions with equal energy (median-split) based on average energy spectral density (across time and subjects). The $N \times N$ matrix $\mathbf{U}^{(\mathrm{low})}$ contains the first $C$ eigenmodes (columns of $\mathbf{U}$) complemented with $N - C$ zero columns. Vice versa, the matrix $\mathbf{U}^{(\mathrm{high})}$ contains the first $C$ zero columns, and then the $N - C$ last eigenmodes. Therefore, filtered signals are obtained following Eqs. (3) and (4):

$$\mathbf{s}_t^\mathrm{C} = \mathbf{U}^{(\mathrm{low})}\mathbf{U}^\mathrm{T}\mathbf{s}_t, \quad (3)$$

$$\mathbf{s}_t^\mathrm{D} = \mathbf{U}^{(\mathrm{high})}\mathbf{U}^\mathrm{T}\mathbf{s}_t. \quad (4)$$

As a measure of structure–function coupling of a specific region, we introduce the structural-decoupling index, i.e. the ratio between the norms of $\mathbf{s}^\mathrm{D}$ and $\mathbf{s}^\mathrm{C}$ across time. For every individual, the maximal excursion under the null, thus over the generated SC-informed surrogates, was used to threshold the structural-decoupling index with a significance level of $\alpha = 1/(19 + 1) = 0.05$. Then, across subjects, the binomial distribution $P(n)$ of having $n$ detections was used to threshold the group average structural-decoupling index, correcting for multiple comparisons for the number of regions tested ($N = 360$).

**Meta-analysis of structural-decoupling index.** A NeuroSynth meta-analysis [https://neurosynth.org/] similar to the one implemented by Margulies et al.[30] was conducted to assess topic terms associated with the structural-decoupling index. Twenty binary masks were obtained by splitting the index values into five-percentile increments and served as input for the meta-analysis, based on the same

24 topic terms adopted by Margulies et al.[30]. Terms were ordered according to the weighted mean of the resulting z-statistics for visualization.

**Reporting summary.** Further information on research design is available in the Nature Research Reporting Summary linked to this article.

## Data availability

The data that support the findings of this study are available in the Human Connectome Project platform (db.humanconnectome.org). Identifiers of the included subjects are reported here: [github.com/gpreti/GSP_StructuralDecouplingIndex]. A reporting summary for this Article is available as a Supplementary Information file. The source data underlying Figs. 2, 3 and Supplementary Figs. 1–5 are provided as a Source Data file.

## Code availability

The code to reproduce the analyses described in this paper is available in the following repository: [https://www.github.com/gpreti/GSP_StructuralDecouplingIndex].

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

## Acknowledgements

This work was supported in part by the Center for Biomedical Imaging (CIBM) of the Geneva-Lausanne Universities and the EPFL, as well as the Leenaards and Louis-Jeantet foundations. Data were provided by the Human Connectome Project, WU-Minn Consortium (Principal Investigators: David Van Essen and Kamil Ugurbil; 1U54MH091657) funded by the 16 NIH Institutes and Centers that support the NIH Blueprint for Neuroscience Research; and by the McDonnell Center for Systems Neuroscience at Washington University.

## Author contributions

M.G.P. and D.V.D.V. designed the experiments, performed the analyses, interpreted the results, and wrote the manuscript.

## Competing interests

The authors declare no competing interests.
