## [Peer Review File · Nature Communications]

Reviewers' Comments:

Reviewer #1:

Remarks to the Author:

This manuscript presents a novel measure of the coupling between brain activity and underlying structural connectivity. Results reveal a gradient in the coupling index from lower-level sensory to higher-level cognitive areas. The methods and results are clearly presented. In addition, the approach makes use of well-motivated null models to test the analyses based on empirical fMRI data. While the manuscript is already polished, I would suggest the authors consider the following points:

- Figure 2: The high-decoupling empirical results presented in (C) appear to preferentially capture regions along the orbitomedial cortex known to be prone to high susceptibility artifacts. While Figure 3 provides excellent evidence that the patterns captured by the structural-decoupling index are broadly more meaningful than reflecting poor SNR, the likely role of noise in the high decoupling of these orbitomedial regions provides an indirect validation that the approach is indeed capturing decoupling (whose functional importance in this specific case may nevertheless be questionable). If the authors agree, such a point may be worth mentioning.

- In the Discussion, the authors point out that cortical thickness also correlates with the patterns observed in their results based on the structural-decoupling index. While there are indeed many reasons for these two measures to be correlated, is it possible that the differences in cortical thickness drive the analysis through improved sampling of gray matter in primary areas. Might this help interpret the divergence of results in the visual and somatosensory/motor cortex presented in Fig. S3?

Minor points:

- Regarding timescales of responses to stimuli, the authors may wish to cite: Chaudhury et al, Neuron, 2015. doi: 10.1016/j.neuron.2015.09.008

- The MRI data analysis was limited to 56 participants out of a possible >1000 from the HCP data. While inclusion of further data is unlikely to alter the results, why did the authors choose to limit their analysis to this relatively small subset? Along those lines, only half of the resting-state fMRI data appears to have been used in the current analysis. Was there a reason for this? The second two sessions of the resting-state fMRI data could provide the basis demonstrating the reliability of their findings. While this may be beyond the scope of the current study, it may be worth considering as additional validation in future work.

Reviewer #2:

Remarks to the Author:

This is an intriguing paper that uses a state-of-the-art harmonic decomposition of the connectome to understand the degree of dependence of functional connectivity on the underlying structural connectome. In particular, the authors introduce and exploit a novel means of generating surrogate functional connectivity data that inherit the zero-th order influence of structural connectivity, but lack more detailed empirical nuances by essentially permuting the data in the (spatial) frequency domain. The authors hence reveal a coupling/dependence gradient which is convergent with recent observations from quite independent techniques.

The idea is excellent and the paper nicely written and illustrated. I think it would appeal to the broad

readership of Nature Communications. I however have some concerns and suggestions as detailed below.

Major:

1. The rationale for the median split in modes (line 104, Figure 1C) for the core coupling/decoupling ratio is not at all clear (*I later saw a heuristic justification in the Supp. Material). This is an essential part of the analysis but is introduced in an ad hoc manner. Better justification could include:
 - 1.1: Fitting the product of 2 Lorentzians to the spectra and using the cross-over/knee (see [M1]);
 - 1.2: performing a formal test of model fit versus model complexity (such as AIC), then truncating the model expansion at the corresponding optimal order (similar to finding the order of a k-means clustering algorithm or an comparable expansion).
2. The authors *could* also consider a surrogate test somewhat orthogonal to the present approach (of permuting functional connectivity) by permuting structural connectivity whilst preserving its geometric embedding [R1], then repeating the (null) analyses (using the empirical functional connectivity). This is just a suggestion but could provide a complimentary and possibly converging perspective.
3. The heuristic explanation of brain activity as a weighted sum of the harmonics seems reasonable (p4), but personally I would also like to this in mathematical form – this could be carefully phrased so as not to dissuade the more general readership. From the description at hand, it is not clear that the activity should be represented as a combination of weighted harmonics (a deterministic component) plus unstructured noise – as I believe it should be (see equation (1) and figure 5 of t1). Perhaps there is a correspondence between the amplitude of this (node-wise) noise term and the authors current measure of structural decoupling..??
4. Line 72: I agree most energy will go into low order modes because that is where the (spatial and temporal) energy is – this really only requires some sort of monotonic drop of power with frequency for the temporal AND spatial spectra – but I don't see strong evidence of a power law (straight line) in the log-log spectra of fig S2 and without a more convincing visual display AND proper inference, I think this claim should be weakened (it's not really essential anyway).
5. To some extent, the authors seem to be generating surrogate data by transforming the data using a spectral decomposition that decorrelates the original dependences and allows random permutation. A brief note connecting this approach to the broader class of resampling null methods that achieve this in the temporal domain with the Fourier or wavelet transform might be warranted for those (like myself) with a long-held interest in this technique 9a brief note could be e.g. added to the Supp. Inf).

Minor:

1. I'm not sure that "structure-function 'coupling' " is the best term, since there are many things (synaptic, neuromodulatory, network) effects that could intervene. I am okay if the authors' wish to keep this term, although more strictly they are quantifying "structure-function 'dependencies' ".
2. L21+: I don't think effective connectivity/DCM is really of relevance to the relationship between structural and functional connectivity unless this is somehow qualified to introduce the notion of task-related modulation and/or the use of structural connectivity priors [S1].
3. L37: In the context of harmonic decompositions of cortex, it is possibly helpful to note that similar decompositions of cortical geometry yield essentially identical outcomes [R2]. The authors *could* also consider this use of such harmonics to predict functional connectivity more formally (and with a clinical application) [T1].
4. Caption, figure 2: "Due to the logarithmic scale, a value of 1 (-1) indicates a double decoupling (coupling) with respect to coupling (decoupling)." –I think this should be better unpacked for the reader.
5. Line 97: I don't think it is "Interesting" that the correlation is stronger with the surrogate than the empirical data, but rather "Reassuring" or "Notable" since it shows that the intended effect of the

surrogate algorithm is indeed observed.

Citations:

F1: Friston, K. J., Harrison, L., & Penny, W. (2003). Dynamic causal modelling. *Neuroimage*, 19(4), 1273-1302.

M1: Miller, K. J., Sorensen, L. B., Ojemann, J. G., & Den Nijs, M. (2009). Power-law scaling in the brain surface electric potential. *PLoS computational biology*, 5(12), e1000609.

S1: Stephan, K. E., Tittgemeyer, M., Knösche, T. R., Moran, R. J., & Friston, K. J. (2009). Tractography-based priors for dynamic causal models. *Neuroimage*, 47(4), 1628-1638.

R1: Roberts, J. A., Perry, A., Lord, A. R., Roberts, G., Mitchell, P. B., Smith, R. E., ... & Breakspear, M. (2016). The contribution of geometry to the human connectome. *Neuroimage*, 124, 379-393.

R2: Robinson, P. A., Zhao, X., Aquino, K. M., Griffiths, J. D., Sarkar, S., & Mehta-Pandey, G. (2016). Eigenmodes of brain activity: Neural field theory predictions and comparison with experiment. *NeuroImage*, 142, 79-98.

T1: Tokariev, A., Roberts, J. A., Zalesky, A., Zhao, X., Vanhatalo, S., Breakspear, M., & Cocchi, L. (2019). Large-scale brain modes reorganize between infant sleep states and carry prognostic information for preterms. *Nature Communications*, 10(1), 2619.

Signed: Michael Breakspear

Manuscript NCOMMS-19-13468 - Response to Reviewers

We thank the two reviewers for their positive and very constructive comments, which led to further considerations by our side on different aspects of the work and to significant improvements of the manuscript. Changes are highlighted in the revised manuscript in blue. In addition, we also provide detailed answers to each comment below.

Reviewers' comments:

Reviewer #1 (Remarks to the Author):

This manuscript presents a novel measure of the coupling between brain activity and underlying structural connectivity. Results reveal a gradient in the coupling index from lower-level sensory to higher-level cognitive areas. The methods and results are clearly presented. In addition, the approach makes use of well-motivated null models to test the analyses based on empirical fMRI data. While the manuscript is already polished, I would suggest the authors consider the following points:

- Figure 2: The high-decoupling empirical results presented in (C) appear to preferentially capture regions along the orbitomedial cortex known to be prone to high susceptibility artifacts. While Figure 3 provides excellent evidence that the patterns captured by the structural-decoupling index are broadly more meaningful than reflecting poor SNR, the likely role of noise in the high decoupling of these orbitomedial regions provides an indirect validation that the approach is indeed capturing decoupling (whose functional importance in this specific case may nevertheless be questionable). If the authors agree, such a point may be worth mentioning.

We thank the reviewer for this useful comment and we agree that potential noise, commonly present in the orbitofrontal regions, would be captured as decoupling. We added a sentence to the discussion to point out this element (lines 211-217). At the same time, in order to give a better visualization of the richness of the structural-decoupling index (SDI) spatial pattern, also pointed out by the reviewer, we changed the brain map in Fig. 3 (left) and reported the binned version of the structural-decoupling index with equally sized nodes, which seems indeed more logical as that is the actual input of the meta-analysis.

- In the Discussion, the authors point out that cortical thickness also correlates with the patterns observed in their results based on the structural-decoupling index. While there are indeed many reasons for these two measures to be correlated, is it possible that the differences in cortical thickness drive the analysis through improved sampling of gray matter in primary areas. Might this help interpret the divergence of results in the visual and somatosensory/motor cortex presented in Fig. S3?

We thank the reviewer for this interesting remark, which led us to deeper thinking on the interpretation of our results in relation to the cortical thickness findings. Although we agree that a poor GM sampling in areas with lower cortical thickness (e.g. visual) might influence the results, possibly leading to lower coupling, we believe that this is not what we see here. In fact, if that would be the case, we would record a lower coupling for visual regions already in the empirical signals (Fig. 2C), while here the

divergence of visual areas does not much concern the empirical SDI values, but rather the surrogates, which show higher coupling in these regions (Fig. 2B). And this ends up in these regions appearing red in Fig. S3, as their structural coupling is higher in SC-informed surrogates than in the empirical case. The pattern of SDI in SC-informed surrogates reflects structural connectivity and partly also known cortical thickness patterns. Visual regions appear in fact to be densely structurally interconnected (areas with highest structural nodal degree) – they were indeed already described as part of the “structural core” (Hagmann et al., 2008)- and characterized by lower cortical thickness, which was reported to be inversely correlated with neural density (Wagstyl et al., 2015).

This said, we think that further (and more specific) analyses would be necessary to be able to expand more on this point, which goes beyond the purpose of this work, but would be of great interest in the future.

References:

- Hagmann et al. doi: [10.1371/journal.pbio.0060159](https://doi.org/10.1371/journal.pbio.0060159)
- Wagstyl et al. doi: [10.1016/j.neuroimage.2015.02.036](https://doi.org/10.1016/j.neuroimage.2015.02.036)

Minor points:

- Regarding timescales of responses to stimuli, the authors may wish to cite: Chaudhury et al, Neuron, 2015. doi: [10.1016/j.neuron.2015.09.008](https://doi.org/10.1016/j.neuron.2015.09.008)

We thank the reviewer for suggestion of this very relevant reference and we included it in the text (ref. 37).

- The MRI data analysis was limited to 56 participants out of a possible >1000 from the HCP data. While inclusion of further data is unlikely to alter the results, why did the authors choose to limit their analysis to this relatively small subset? Along those lines, only half of the resting-state fMRI data appears to have been used in the current analysis. Was there a reason for this? The second two sessions of the resting-state fMRI data could provide the basis demonstrating the reliability of their findings. While this may be beyond the scope of the current study, it may be worth considering as additional validation in future work.

The selection of the sample was mainly limited by the structural connectivity computation time (multi-shell multi-tissue spherical deconvolution and whole-brain probabilistic tractography). Given that only group results were reported (no use of subject-specific features or traits) and that findings were replicable across different subsamples (smaller preliminary sample of 21 acquisitions, then bigger ones of 56 and 112), we considered this as an acceptable sample size for an fMRI study. However, we definitely agree with the reviewer that the inclusion of the second two resting-state sessions would be very helpful for validation of the results. Therefore, we performed the same analysis on these additional datasets and we compared the results. Findings were highly reliable across the test-retest sessions (correlation between SDI patterns –in empirical case and surrogates- always >0.9, similar behavioral characterization from the meta-analysis). We describe this in the section “Reliability across sessions” in the Results (lines 142-150), and we added sentences to the Introduction (lines 58-59) and Methods (lines 244-246).

Reviewer #2 (Remarks to the Author):

This is an intriguing paper that uses a state-of-the-art harmonic decomposition of the connectome to understand the degree of dependence of functional connectivity on the underlying structural connectome. In particular, the authors introduce and exploit a novel means of generating surrogate functional connectivity data that inherit the zero-th order influence of structural connectivity, but lack more detailed empirical nuances by essentially permuting the data in the (spatial) frequency domain. The authors hence reveal a coupling/dependence gradient which is convergent with recent observations from quite independent techniques.

The idea is excellent and the paper nicely written and illustrated. I think it would appeal to the broad readership of Nature Communications. I however have some concerns and suggestions as detailed below.

Major:

1. The rationale for the median split in modes (line 104, Figure 1C) for the core coupling/decoupling ratio is not at all clear (*I later saw a heuristic justification in the Supp. Material). This is an essential part of the analysis but is introduced in an ad hoc manner. Better justification could include:

1.1: Fitting the product of 2 Lorentzians to the spectra and using the cross-over/knee (see [M1]),

1.2: performing a formal test of model fit versus model complexity (such as AIC), then truncating the model expansion at the corresponding optimal order (similar to finding the order of a k-means clustering algorithm or an comparable expansion).

We thank the reviewer for pointing out this lack of clarity in the manuscript. In fact, the motivation behind the application of the median split criterium to the PSD was to equally divide the **signal** energy into two (equal) portions. It was therefore not our intention to use the knee in the PSD distribution as cut-off frequency: as we have increasing amount of noise with higher frequencies, we would include too much noise vs. signal in the high-frequency (decoupled) portion by increasing the cut-off (now at $\Lambda = 0.35$, under the median split criterium).

However, it is true that we would like the median split criterium to be applied only to the signal, excluding the contribution of the noise, which is in principle equally distributed in all frequencies. We see the point in trying to propose a model considering a noise component (also referring to the reviewer's comment 3), as this could allow us to remove a bias due to the noise, that might change our cutoff. To this purpose, we fitted the PSD with two distributions to segregate the signal and noise contributions; i.e., the product of 2 Lorentzians, as suggested by the reviewer and detailed in [M1] + one constant for the noise. However, the estimate that we obtain for the noise is 0, so no change in the cutoff was applied. We can assume therefore that all contributions to the energy density are signal relevant, which is what we do when applying the GFT and considering all harmonics. We added a sentence to the Methods to clarify this (lines 277-278).

2. The authors *could* also consider a surrogate test somewhat orthogonal to the present approach (of permuting functional connectivity) by **permuting structural**

connectivity whilst preserving its geometric embedding [R1], then repeating the (null) analyses (using the empirical functional connectivity). This is just a suggestion but could provide a complimentary and possibly converging perspective.

We thank the reviewer for the constructive suggestion. Indeed, we agree that the generation of matrices which randomize structural connectivity while preserving geometric embedding could offer a complementary perspective, and a possible alternative to create surrogates. Conceptually, we see this option as in-between our SC-ignorant surrogates (where only the nodal strengths -but not the geometry- is preserved) and the SC-informed surrogates (where the whole SC information is kept). We applied the same pipeline to our data, but we see a very different distribution of weights vs fiber lengths (reported in the following plot), possibly due to the different way used to construct the connectome, for which the whole procedure should be adapted. We therefore decided not to pursue it in the paper, but we comment on it in the Discussion (lines 208-210).

3. The heuristic explanation of brain activity as a weighted sum of the harmonics seems reasonable (p4), but personally I would also like to this **in mathematical form** – this could be carefully phrased so as not to dissuade the more general readership. From the description at hand, it is not clear that the activity should be represented as a combination of weighted harmonics (a deterministic component) **plus unstructured noise** – as I believe it should be (see equation (1) and figure 5 of t1). Perhaps there is a correspondence between the amplitude of this (node-wise) noise term and the authors current measure of structural decoupling..??

We thank the reviewer for these insightful considerations. In this case, we are not proposing a model but rather an algorithm that allows to decompose the functional signal into harmonics. The mathematical form describing this decomposition corresponds to the inverse graph Fourier transform: $st = U*s_hat$ (Methods section P.16, line 277). As we consider the full decomposition (all eigenvectors u_k), this already includes unstructured noise, in principle equally distributed in all frequencies, without the need of a separate noise term. However, it's true that we have so far ignored the contribution of noise to our coupled/decoupled reconstructed signals, and, as suggested by the reviewer, this could influence the decoupling measure. In fact, assuming equal contribution of noise to each frequency, a bigger amount of noise will be included in the decoupled part of the signal (where we include 339 components vs. 21 in the coupled). However, it is not trivial to estimate the noise component and, as showed in the response to comment 1, the resulting noise is 0.

We further assume then that no considerable bias affects the SDI calculation. We added a comment on this in the Discussion (lines 211-213).

4. Line 72: I agree most energy will go into low order modes because that is where the (spatial and temporal) energy is – this really only requires some sort of monotonic drop of power with frequency for the temporal AND spatial spectra – but I don't see strong evidence of a power law (straight line) in the log-log spectra of fig S2 and without a more convincing visual display AND proper inference, I think this claim should be weakened (it's not really essential anyway).

We agree with the reviewer that the power law description is not essential. We changed the text accordingly to weaken this claim and simply highlight the domination of LF energy.

5. To some extent, the authors seem to be generating surrogate data by transforming the data using a spectral decomposition that decorrelates the original dependences and allows random permutation. A **brief note connecting** this approach to the broader class of **resampling null methods** that achieve this in the temporal domain with the Fourier or wavelet transform might be warranted for those (like myself) with a long-held interest in this technique (a brief note could be e.g. added to the Supp. Inf).

We thank the reviewer for this relevant suggestion and included a brief note on these approaches (lines 89-92).

Minor:

1. I'm not sure that "structure-function 'coupling' " is the best term, since there are many things (synaptic, neuromodulatory, network) effects that could intervene. I am okay if the authors' wish to keep this term, although more strictly they are quantifying "structure-function 'dependencies' ".

We thank the reviewer for this suggestion. We would like to keep the term *coupling*, (we agree on *dependence*, but we think it might recall too much correlational approaches), but we also think that it might generate some confusion. To avoid that, we added a sentence to the Introduction to better clarify this and reduce misunderstanding (lines 40-42).

2. L21+: I don't think effective connectivity/DCM is really of relevance to the relationship between structural and functional connectivity unless this is somehow qualified to introduce the notion of task-related modulation and/or the use of structural connectivity priors [S1].

We agree with the reviewer and we added the suggested concept/reference (line 24, ref. 5).

3. L37: In the context of harmonic decompositions of cortex, it is possibly helpful to note that similar decompositions of cortical geometry yield essentially identical

outcomes [R2]. The authors *could* also consider this use of such harmonics to predict functional connectivity more formally (and with a clinical application) [T1].

We thank the reviewer for the suggestion and we added a comment on that (lines 70-72, refs. 24-25).

4. Caption, figure 2: “Due to the logarithmic scale, a value of 1 (–1) indicates a double decoupling (coupling) with respect to coupling (decoupling).” –I think this should be better unpacked for the reader.

We expanded that caption part to make it more clear for the reader.

5. Line 97: I don’t think it is “Interesting” that the correlation is stronger with the surrogate than the empirical data, but rather “Reassuring” or “Notable” since it shows that the intended effect of the surrogate algorithm is indeed observed.

We definitely agree and we changed the text accordingly.

Citations:

F1: Friston, K. J., Harrison, L., & Penny, W. (2003). Dynamic causal modelling. *Neuroimage*, 19(4), 1273-1302.

M1: Miller, K. J., Sorensen, L. B., Ojemann, J. G., & Den Nijs, M. (2009). Power-law scaling in the brain surface electric potential. *PLoS computational biology*, 5(12), e1000609.

S1: Stephan, K. E., Tittgemeyer, M., Knösche, T. R., Moran, R. J., & Friston, K. J. (2009). Tractography-based priors for dynamic causal models. *Neuroimage*, 47(4), 1628-1638.

R1: Roberts, J. A., Perry, A., Lord, A. R., Roberts, G., Mitchell, P. B., Smith, R. E., ... & Breakspear, M. (2016). The contribution of geometry to the human connectome. *Neuroimage*, 124, 379-393.

R2: Robinson, P. A., Zhao, X., Aquino, K. M., Griffiths, J. D., Sarkar, S., & Mehta-Pandey, G. (2016). Eigenmodes of brain activity: Neural field theory predictions and comparison with experiment. *NeuroImage*, 142, 79-98.

T1: Tokariev, A., Roberts, J. A., Zalesky, A., Zhao, X., Vanhatalo, S., Breakspear, M., & Cocchi, L. (2019). Large-scale brain modes reorganize between infant sleep states and carry prognostic information for preterms. *Nature Communications*, 10(1), 2619.

Signed: Michael Breakspear

Reviewers' Comments:

Reviewer #1:

Remarks to the Author:

I would like to thank the authors for thoroughly addressing my prior comments in their revised manuscript.

Reviewer #2:

Remarks to the Author:

Thank you for your responses to my prior concerns.

The figure in your response to my 2nd reply suggests there are a lot of very weak spurious connections in the connectome but as these are several orders of magnitude than the stronger distance dependent ones, they are likely to be negligible in any case.

Manuscript NCOMMS-19-13468A - Response to Reviewers

We thank the two reviewers for their final positive decision.

Reviewers' comments:

Reviewer #1 (Remarks to the Author):

I would like to thank the authors for thoroughly addressing my prior comments in their revised manuscript.

The reviewer does not require any further action from the authors.

Reviewer #2 (Remarks to the Author):

Thank you for your responses to my prior concerns.

The figure in your response to my 2nd reply suggests there are a lot of very weak spurious connections in the connectome but as these are several orders of magnitude than the stronger distance dependent ones, they are likely to be negligible in any case.

We agree with the reviewer on this interpretation. Indeed the shape of the distribution is more similar to the referenced paper when deleting these weak connections. The reviewer does not require any further action from the authors.